# Molecular Mapping to Discover Reliable Salinity-Resilient QTLs from the Novel Landrace Akundi in Two Bi-Parental Populations Using SNP-Based Genome-Wide Analysis in Rice

**DOI:** 10.3390/ijms241311141

**Published:** 2023-07-06

**Authors:** Sheikh Maniruzzaman, M. Akhlasur Rahman, Mehfuz Hasan, Mohammad Golam Rasul, Abul Hossain Molla, Hasina Khatun, K. M. Iftekharuddaula, Md. Shahjahan Kabir, Salma Akter

**Affiliations:** 1Plant Breeding Division, Bangladesh Rice Research Institute (BRRI), Gazipur 1701, Bangladesh; skmonir85@yahoo.com (S.M.);; 2Department of Genetics and Plant Breeding, Bangabandhu Sheikh Mujibur Rahman Agricultural University, Gazipur 1706, Bangladesh; 3Department of Environmental Science, Bangabandhu Sheikh Mujibur Rahman Agricultural University, Gazipur 1706, Bangladesh; ahmolla@bsmrau.edu.bd; 4Plant Physiology Division, Bangladesh Rice Research Institute (BRRI), Gazipur 1701, Bangladesh

**Keywords:** genetic loci, salinity-resilient QTLs (SRQ), salt-tolerant indica germplasm (Akundi), single nucleotide polymorphisms (SNP), 1k-RiCA, *Oryza sativa* L.

## Abstract

Achieving high-yield potential is always the ultimate objective of any breeding program. However, various abiotic stresses such as salinity, drought, cold, flood, and heat hampered rice productivity tremendously. Salinity is one of the most important abiotic stresses that adversely affect rice grain yield. The present investigation was undertaken to dissect new genetic loci, which are responsible for salt tolerance at the early seedling stage in rice. A bi-parental mapping population (F_2:3_) was developed from the cross between BRRI dhan28/Akundi, where BRRI dhan28 (BR28) is a salt-sensitive irrigated (*boro*) rice mega variety and Akundi is a highly salinity-tolerant Bangladeshi origin *indica* rice landrace that is utilized as a donor parent. We report reliable and stable QTLs for salt tolerance from a common donor (Akundi) irrespective of two different genetic backgrounds (BRRI dhan49/Akundi and BRRI dhan28/Akundi). A robust 1k-Rice Custom Amplicon (1k-RiCA) SNP marker genotyping platform was used for genome-wide analysis of this bi-parental population. After eliminating markers with high segregation distortion, 886 polymorphic SNPs built a genetic linkage map covering 1526.5 cM of whole rice genome with an average SNP density of 1.72 cM for the 12 genetic linkage groups. A total of 12 QTLs for nine different salt tolerance-related traits were identified using QGene and inclusive composite interval mapping of additive and dominant QTL (ICIM-ADD) under salt stress on seven different chromosomes. All of these 12 new QTLs were found to be unique, as no other map from the previous study has reported these QTLs in the similar chromosomal location and found them different from extensively studied *Saltol*, *SKC1*, *OsSalT*, and sal*T* locus. Twenty-eight significant digenic/epistatic interactions were identified between chromosomal regions linked to or unlinked to QTLs. Akundi acts like a new alternate donor source of salt tolerance except for other usually known donors such as Nona Bokra, Pokkali, Capsule, and Hasawi used in salt tolerance genetic analysis and breeding programs worldwide, including Bangladesh. Integration of the seven novel, reliable, stable, and background independent salinity-resilient QTLs (*qSES1*, *qSL1*, *qRL1*, *qSUR1*, *qSL8*, *qK8*, *qK1*) reported in this investigation will expedite the cultivar development that is highly tolerant to salt stress.

## 1. Introduction

Rice (*Oryza sativa* L. 2n = 2x = 24) is a diploid (2n = 2x = 24) glycophyte. As a genetic model crop, it was completely sequenced first in *Oryza sativa* ssp. *Indica* cv 93-11 during the mid-2000s [1] and in *Oryza sativa* ssp. *Japonica* cv Nipponbare cultivars and annotations of gene sequences [2] because of its comparatively small genome size and enormous genetic diversity. Moreover, this is the staple food crop in the world, which is consumed by more than half of the world’s population [3,4,5], who directly depend on it for their source of nutrition [6]. It is also an intrinsic part of our rituals and rites in South and Southeast Asia. Thus, rice is not only important as a food crop or agricultural commodity worldwide, but also as an area of extensive study in the field of genetics, genomics, biotechnology, and agronomy including international agriculture.

Several abiotic stress-related problems affect rice production, and salinity is one of them. Salinity is a key abiotic stress in this changed climatic condition that affects approximately one-third of land areas of the world. Globally, salinity stress poses enormous challenges to food security [7]. Because the rice plant is basically sensitive to salt stress, increasing the level of salinity detrimentally affects productivity [8]. The extent of saline-prone vulnerable areas will continue to expand in these altered climatic conditions [8]. High salinity impedes the water and nutrient uptake from soil, and this inhibits seedling growth and development and hence reduces production. Consequently, breeding programs targeting coastal saline zones must emphasize salt-stress-tolerant rice cultivar development. 

An increase in rice productivity in coastal saline-affected regions is inevitable [9]. Salinity stress is a severe constraint on coastal rice farming, owing to its susceptibility during the young seedling stage and reproductive development (panicle initiation to flowering stages) [10]. Various studies have demonstrated that salt tolerance is influenced by polygenes and QTL [11]. By applying QTL dissection in segregating populations, several genetic loci for different traits related to salinity tolerance in rice have been reported. Hossain et al. [12] stated that by using 131 microsatellite markers in an F_2_ biparental mapping population, sixteen QTLs with LOD values ranging from 3.2 to 22.3 were mapped on chromosomes 1, 7, 8, and 10. Rahman et al. [11] observed that genomic loci on the bottom of the long arm of chromosome 1 were responsible for Na^+^ concentration, K^+^ uptake, Na^+^/K^+^ ratio, and % survival; chromosome 3 was responsible for less Na^+^ accumulation, maintaining lower Na^+^/K^+^ ratio, ability of Na^+^ removal from leaf tissues, and overall phenotypic score applying the standard evaluation system (*SES*) score (for scoring of visual salt injury at seedling and reproductive stages in rice); and QTL located on chromosome 5 was controlled SES. Recently, Goto et al. [13] revealed that excess buildup of salt in rice plants under salt stress reduces growth, development, and grain yield. Salt exclusion ability in leaves is an important tolerance mechanism to decrease salt influx and accumulation in the leaves that maintain photosynthesis under salt stress. They identified QTLs using the F_2_ population derived from two rice varieties: IR-44595 with superior Na^+^ exclusion ability, and 318 with contrasting traits. However, the QTL reported on chromosome 11 co-occurred with other QTLs controlling Na^+^ concentration in shoots, leaf blades, and leaf sheaths, and Na^+^/K^+^ ratio in leaf blades. Additionally, six pairwise significant digenic relationships between QTL-linked and QTL-unlinked (QTL vs. background loci) regions were detected. Nayyeripasand et al. [14] postulated 151 trait marker associations involved in stress resistance on rice chromosome 10 that were arranged in 29 genomic areas.

Investigating effective traits in rice germplasm is necessary for detecting desirable alleles. However, Akundi is a salt-tolerant, Bangladeshi origin indica landrace is considerably acclimated, deposited less Na^+^ and proportionally more K^+^ in shoots/leaves, and maintained a minimal Na^+^/K^+^ ratio. They considerably restrict sodium translocation from roots to the shoots, which was known as a useful and novel salinity tolerance source [15]. Well-recognized salinity-tolerant donors are Pokkali, Capsule, Hasawi, Nona Bokra, Cheriviruppu, but ‘Akundi’ serve as substitute new donors of salt tolerance for extensive genetic investigation and breeding programs globally including Bangladesh. Pokkali shows better adaptability to salt stress because of two different features to cope with difficult salt stress conditions: it’s capacity to keep a less Na^+^/K^+^ ratio to maintain ionic equilibrium in the shoot/leaf tissue, consequently it maintains vigorous growth rate in saline ecosystems, which helps in diffusion of the salt and reducing the stress-related cytotoxicity within the plant tissue [16,17]. An F_2:3_ biparental mapping population was generated from a cross of BRRI dhan28 and Akundi, where BRRI dhan28, a sensitive irrigated (*boro*) season rice variety, was crossed with Akundi, a salt-tolerant traditional variety. The mapping of genetic loci is important for enhancing and deciphering our insights on the type of the distribution, genetic, and genomic architecture of component characters of interest [18,19]. In addition, it enables the development of markers for complicated quantitative traits and useful alleles in breeding programs. The main objective of the present investigation is to dissect new and stable quantitative trait loci (QTLs) from common salt-tolerant parent (donor) Akundi using two different sensitive parents, BRRI dhan49 and BRRI dhan28, which control salt tolerance in the early seedling stage of rice to find epistatic/diallelic interactions.

## 2. Results

### 2.1. Salt Stress Reactions of the Selected F_2:3_ Progenies and the Parental Genotypes

Ninety-two F_2:3_ individuals were obtained from BRRI dhan28/Akundi cross, where BRRI dhan28 was a sensitive variety and Akundi was salt-tolerant and were assessed in phytotron using a hydroponic system under a salt stress condition. The F_2_ population is adequately dissimilar, indicating that these individuals have significant genetic variability from one another, as explained further below.

### 2.2. Assessing Agronomic Characters under Salt Stress

#### 2.2.1. SES Score

Two parents showed significantly different reactions to salt stress, where BRRI dhan28 (BR28) was found to be highly sensitive (SES score = 8) and Akundi demonstrated tolerance to salinity (SES score = 3). The distribution of this trait is likely multifactorial and negatively skewed (skewness = −0.74). Figure 1a illustrates a histogram that shows the frequency of magnitude of different variables/traits of the mapping population and its parents. The ultimate visual SES score for the tolerant plants ranged from 4 to 5, with an average of 4.5, while the SES score for the sensitive individuals ranged from 6 to 8 (average SES score = 7.0).

#### 2.2.2. Survival

The average % survival for Akundi was 100%, while the average survival of BR28 was 45% (range: 40–50%). This trait showed a negative skew (skewness = −1.37). A higher rate of survival (90–100%) was observed in about 79% of individuals in the biparental population (Figure 1b).

#### 2.2.3. Shoot Length

The average length of the shoot was 57.2 cm in tolerant parent Akundi (range: 55.8–58.6 cm), while BR28 had a mean shoot length of 28.00 cm (range: 27.0–29.0 cm). Significant variability was observed for shoot length among F_2:3_ progenies (range: 30.85–55.31 cm). Moreover, the frequency distribution reveals that the distribution is fairly symmetrical with a skewness of 0.24 (Figure 1c).

#### 2.2.4. Shoot Dry Weight

The shoot dry weight ranged between 0.05 and 0.16 g, with shoot dry weight obtained at 0.21 g in Akundi and 0.08 g in BR28 (Figure 1d). A total of 31 out of the 92 F_2:3_ progenies had shoot biomass in the range of 0.12–0.20 g, while only 13 had shoot biomass of less than 0.08 g, comparable to that of BR28; this distribution is skewed negatively (−0.14).

#### 2.2.5. Root Length

Average root length of Akundi was 13.3 cm (range of 13.2–13.4 cm), while the mean root length of BR28 was 12.0 cm (range of 11.5–12.5 cm) (Figure 1e). In root length, large variation was found in the F_2:3_ individuals (range of 8.35–18.17 cm). The frequency distribution is negatively skewed (−0.25), and the distribution is slightly symmetrical.

### 2.3. Characterizing Physiological Parameters under Salt Stress

#### 2.3.1. SPAD Value

SPAD value ranged from 14.5 to 30.8. The SPAD value of Akundi was 29.70 and BR28 was 22.65 (Figure 1f). The highest SPAD value was more than 29.70, observed in 22 individuals, and the lowest SPAD value was less than 20.00, found in 8 plants. The data were negatively skewed (−0.15).

#### 2.3.2. Na^+^ Concentration

Na^+^ concentration was less than 0.1 mmolg^−1^ dwt in around 33 individuals; 59 plants had higher Na^+^ concentration (more than 0.10 mmolg^−1^ dwt). The Na^+^ concentration in tolerant parent Akundi (0.18 mmolg^−1^ dwt) and sensitive parent BR28 (0.31 mmolg^−1^ dwt) vary considerably. The data shows positive skewness (0.21) (Figure 1g).

#### 2.3.3. K^+^ Concentration

The K^+^ concentration was positively skewed (0.35). Around 13 plants had low (less than 0.47 mmog^−1^ dwt) K^+^ concentration, 30 plants had high (more than 0.70 mmolg^−1^ dwt) K^+^ concentration, and the rest of the individuals had an intermediate K^+^ concentration. The potassium concentration of Akundi (1.12 mmolg^−1^ dwt) and BR28 (0.47 mmolg^−1^ dwt) exhibit significant differences (Figure 1h).

#### 2.3.4. Na^+^/K^+^ Ratio

The Na^+^/K^+^ ratio of tolerant Akundi and susceptible parent BR28 had 0.16 and 0.34, respectively. The Na^+^/K^+^ ratio of around 30 progenies had low (less than 0.16 mmolg^−1^ dwt) and 13 plants had high (more than 0.30 mmolg^−1^ dwt) Na^+^/K^+^ ratio. The frequency distribution is positively skewed, and the value of skewness was 0.84 (Figure 1i).

### 2.4. Trait Correlation Analysis between Different Characters

For polygenic traits of salinity, efficient selection strategies rely on the information on the association between overall phenotypic performance/salt injury scores (SES Score) and their contributing characteristics. Positive correlations were observed among salt injury scores (SES), shoot length (cm), shoot dry weight (g), and Na^+^ concentration (mmolg^−1^ dwt), and significant and negative correlations with K^+^ concentration, and SPAD value (Figure 2). Here, positive correlation was found between SES and Na^+^ concentration. If the Na^+^ concentration increases in the plant tissue, then the visual salt injury scores (SES score) also increase; that is, the two traits have a cause–effect relationship and are reliant on each other. The correlation of SES with K^+^ concentration and SPAD value has a negative association, demonstrating that the visual symptom (SES) decreases as the accumulation of chlorophyll and potassium concentration increases in the plant tissue. The % Survival revealed a significant and positive correlation with shoot length and K^+^ concentration. Therefore, a high amount of potassium and low SES score are important considerations for plant survival under salt stress. Potassium (K^+^) concentration has also a significant and positive correlation with % survival, shoot length (cm), shoot dry weight (g), and root length (cm), and a negative correlation was observed with the SES score. Consequently, K^+^ concentration is a key parameter in rice plants through maintaining ionic balance (homeostasis) under salt stress conditions (Figure 2).

### 2.5. Determining the Contribution of Component Agronomic and Physiological Traits (Independent Variables) to Overall Phenotypic Performance (SES Score: Dependent Variable) through Path Analysis

Overall phenotypic performance (SES score) under salt stress depends on the contribution of various agronomic and physiological traits under study. A higher magnitude of the SES Score indicates a lower level of salt tolerance, and vice versa. The Na^+^ concentration and Na^+^/K^+^ ratio had a significant positive effect on overall phenotypic performance (SES), whereas a negative direct effect was observed between SES score and shoot dry weight, root length, SPAD value, and K^+^ concentration (Table 1). It is indeed the circumstance that a lower SES score shows more tolerance to salinity and less amount of Na^+^ concentration, and the Na^+^/K^+^ ratio corresponds to a lower SES score, which shows consistency with the correlation (0.24 and 0.33). Na^+^ concentration had an indirect effect on tolerance via root length. It demonstrates that the direct negative effect (−0.16) of root length on SES is indicative of tolerance. The values of other traits such as % survival, shoot length, shoot dry weight, SPAD value, K^+^ concentration, and Na^+^/K^+^ ratio was used to calculate indirect path coefficients, indicating the extent to which this trait indirectly influenced the SES score through their effects on other traits (Table 1). The overall magnitude of indirect effects was estimated by adding the indirect path coefficients for each character.

A residual effect (R) is 0.27; this means that the variables % survival, shoot and root length, shoot dry weight, SPAD value, sodium (Na^+^) concentration, potassium (K^+^) concentration, and Na^+^/K^+^ ratio are combinedly responsible for 73% of the phenotypic variation in salinity tolerance.

### 2.6. SNP Marker Polymorphism and Construction of Genetic Linkage Map

Marker polymorphism of 1k-RiCA SNP markers was checked between two parents (BR28 vs. Akundi). In total, 886 polymorphic markers (88.6% polymorphism) were identified (Appendix A) and they showed differences (polymorphism) from each other for both parents, and these polymorphic SNP markers were then used for genotyping the 92 F_2_ individuals. Polymorphic SNPs were sorted on excel sheets based on the first chromosome number and then their physical distance in ascending order. A molecular map of QTL regions was constructed using QTL IciMapping Version 4.2 software. The linkage map was approximately 1526.5 cM in length [20], with a mean interval length of 1.72 cM (Figure 3). At a threshold LOD of 3.0, 12 QTLs were identified applying interval mapping and composite interval mapping (Table 2). Robust QTLs were shown on the genomic linkage map. Figure 4 illustrates the QTL likelihood (LOD curve) graphs for newly detected QTL loci controlling traits linked with seedling stage salt tolerance.

### 2.7. Salinity-Tolerant QTLs Controlling Agronomic Traits

One QTL, *qSES1*, was reported for visual/phenotypic symptoms using SES score, which was responsible for 15.6% phenotypic variance. Here, the QTL position was 151.8 cM, and the QTL contributing parent is Akundi. The Akundi allele (*qSES1*) significantly enhanced salt tolerance ability through reducing SES visual score. On chromosome 11, one major QTL (*qSUR11*) for survival was revealed. For the QTL *qSUR11*, the phenotypic variance was 16.1%, which is contributed by BR28. Two QTLs were identified on chromosomes 1 and 8 that were significantly associated with shoot length nearby the QSES1-2_2 and GM4_4; here, the R^2^ value is 30.7% and 16.3%, respectively, by the interval mapping (IM) and composite interval mapping (CIM) method. Two QTLs, *qSDW1* and *qSDW10*, were observed on chromosomes 1 and 10, of which phenotypic variations were 16.4% and 17.1%, respectively. Here, the QTL positions were 123.4 and 68.6 cM and the locus names chro01_231396842 and chr10_17397576 contributed by the allele was Akundi. The QTL *qRL1* was detected from the choromosome1 by single marker regression method (LOD value for SES score trait 3.4, survival LOD: 3.5 and LOD for shoot length is 7.3, LOD for shoot dry weight 3.6 and root length LOD:3.5 in QGene program) (Table 2).

### 2.8. QTL Regions Governing Physiological Characters

QTL related to chlorophyll content was identified using SPAD readings, with a significant LOD value. This QTL is located at the 91.4 cM position on chromosome 7, which explained 12.1% phenotypic variation. Two QTLs (*qNa2* and *qNa10*) were identified on chromosomes 2 and 10, respectively, which are significantly linked with the Na^+^ concentration trait in plant tissue by interval mapping and composite interval mapping method. QTL *qNa2* and *qNa10* accounted for about 14.5% and 15.5% of the total phenotypic variation, respectively, and exhibited salt tolerance through reducing 1.43 mmolg^−1^ dwt and 0.74 mmolg^−1^ dwt Na^+^ content in plant tissue, respectively (Table 2). One significant QTL (*qK1*) was identified through single marker regression (SMR) for K^+^ concentration. These QTLs are found on chromosome 1, accounting for about 19.3% of the total phenotypic variation in K^+^ concentration. The QTL of *qK1* was contributed by BR28. One QTL (*qNaKR12*) was detected for Na^+^/K^+^ ratio, which is located on chromosome 12, and the locus name is chr12_10051752. The position of the QTL is 39.4 cM, whereas the additive effect is −0.04 and 12.1% of the total variation (SPAD readings LOD: 3.0, Na^+^ concentration LOD: 3.3, LOD: 3.0 for K^+^ concentration and Na^+/^K^+^ ratio LOD: 3.0 analyzed using QGene) (Table 2).

### 2.9. Probable Functional Genes Detection in the Different QTL Regions

The QTLs controlling traits such as SES score (*qSES1*) were observed in the chromosomal region 38723347–38724165 bp on chromosome 1 with 16 functional genes. The putative function of the candidate gene (LOC_Os01g66670) was expressed protein and drought-induced proteins, anther, and pollen wall remodeling/metabolism proteins contributing to the salt stress tolerance of rice. The QTLs for shoot length (*qSL1*), shoot dry weight (*qSDW1*), root length (*qRL1*), and K^+^ concentration (*qK1*) were also found in chromosome 1. However, the QTL position, number of loci, candidate gene, and putative function are different from one another. The position of QTL, locus number, candidate gene, and putative function of shoot length (*qSL1*) was 39,794,226–39,799,341 bp, 18, LOC_Os01g68490 and tetra-trico peptide-like helical, putative, expressed, abscisic acid responses and tolerance to osmotic stress, enable plants to acclimatize with adverse environmental conditions, respectively. The QTL position, number of loci, candidate gene, and putative function of shoot dry weight (*qSDW1*) was 31,473,897–31,477,599 bp, 17 loci, LOC_Os01g54700 and retrotransposon protein, putative, Ty1-copia subclass, expressed, regulating gene expression during the development of plant under salt stress, played a major role in shaping genome structure, respectively. The position of QTL, number of loci, candidate gene, and putative function of root length (*qRL1*) was 10,713,139–10,714,271 bp, 16 loci, LOC_Os01g18950 and peroxidase precursor, putative, expressed, increases defense against oxidative stress, highly tolerant to different stresses allowing survival when the water supply is a limiting factor, respectively. The QTL position of *qK1* for K^+^ concentration, total number of loci, candidate gene, and putative function was 39922192–39923794 bp, 16, LOC_Os01g68730 and RNA-binding protein FUS, putative, expressed, involved in cellular stress response, responses against pathogen infection, respectively. Another functional gene, LOC_Os11g09990, was observed on chromosome 11 encoding mTERF family protein, expressed, coordinate mitochondrial transcription, growth, development, and stress response between the chromosomal region 5,357,710–5,359,379 bp inside the QTL *qSUR11* at the seedling stage. In the chromosomal region 5,613,578–5,616,235 bp of chromosome 8, one functional gene LOC_Os08g09715 was identified as F-box domain-containing protein expressed; F-box protein-encoding genes during the floral transition as well as panicle and seed development play a variety of roles in the developmental processes including plant hormonal signal transduction, floral development, secondary metabolism, senescence, circadian rhythms, and responses to both biotic and abiotic stresses within the significant QTLs, *qSL8* of shoot length at the seedling stage. The QTLs of shoot dry weight (*qSDW10*) and Na^+^ concentration (*qNa10*) were found in chromosome 10. However, their QTL position, number of loci, candidate gene, and putative function are different from one another. The QTL position, number of loci, candidate gene, and putative function of shoot dry weight (*qSDW10*) and Na^+^ concentration (*qNa10*) were 17,547,961–17,548,260 bp, 11 loci, LOC_Os10g33360, expressed protein, drought-induced proteins, anther and pollen wall remodeling/metabolism proteins contribute to the tolerance of rice to salt stress, respectively. SPAD value (*qSPAD7*), Na^+^ concentration (*qNa2*), and Na^+^/K^+^ ratio (*qNaK12*) were also found in chromosome in 7, 2 and 12 in QTL position 23,314,482–23,319,154 bp, 34,274,401–34,276,956 bp, and 10,048,848–10,053,017 bp, respectively. Here, the candidate gene and putative function were LOC_Os07g38860, LOC_Os02g56010, LOC_Os12g17530 and OsGH3.10—Probable indole-3-acetic acid-amidosynthetase, expressed; anthocyanidin 3-O-glucosyltransferase, putative, expressed and expressed protein, drought-induced proteins, anther and pollen wall remodeling/metabolism proteins contribute to the tolerance of rice to salt stress respectively at the seedling stage (Appendix A).

### 2.10. Epistatic Interaction

Epistasis is important to control quantitative characters by maintaining interactions between two alleles at a number of loci. For all traits, a two-way test was performed to identify three types of interactions, such as (i) interaction between complementary loci, (ii) interaction between QTLs, and (iii) interaction between QTLs and background loci using the ICIM-EPI method from QTL IciMapping Version 4.2 software. The epistatic analysis revealed 28 significant interactions. They were made up of eight interactions for SES score, two marker intervals (MI) for % survival, five marker loci intervals for shoot length, one interaction for root length, shoot dry weight, and Na^+^ and K^+^ concentration. Five marker intervals for SPAD value and four intervals for Na^+^/K^+^ ratio were spread across eleven different chromosomes (1, 2, 3, 4, 5, 6, 7, 9, 10, 11, and 12) (Appendix A). Two types of digenic interactions (complementary type and between QTLs background type interaction) were identified (Appendix A; Figure 5). (I) interaction between the QTL (marker interval M180-M181; *qSDW10*) on chromosome 10 for shoot dry weight and background loci (such as marker interval M510-M511) on chromosome 6 with LOD of 34 for shoot dry weight (Appendix A; Figure 5a) and (II) interaction between complementary locus (90 cM; MI: M165-M166) on chromosome 2 and 61 cM; MI: M769-M770 on chromosome 10 for SPAD value (Figure 5b). Interaction between background loci at MI: M247-M248 (62 cM) on chromosome 3 and MI: M921-M922 on chromosome 12 (76 cM) with LOD of 48.7 and PVE of 0.50% for SES score (Figure 5c). Interaction between QTLs is not observed in this study. Two marker pairs significantly affected the phenotypic expression of the trait through the interaction between the QTL and background loci and 26 significant interactions between complementary loci, thus demonstrating robust interaction effects. Out of two QTLs and background, epistasis interaction for the shoot dry weight trait on chromosomes 6 and 10 had a high LOD value (34) with PVE 1.33%. Out of 26 complementary loci, a high LOD value (69) was observed in the survival percentage with 1.32% phenotypic variation explained.

### 2.11. Stable QTLs for Different Agronomic and Physiological Traits Responsible for Salinity Tolerance

Fifteen QTLs reported the mapping population derived from BR49/Akundi [7], and 12 QTLs controlling different salt-tolerant traits were identified from the present investigation using the population from the BR28/Akundi cross. A total of seven genetic loci (*qSES1*, *qSL1*, *qRL1*, *qSUR1*, *qSL8*, *qK8*, *qK1*) were found to be stable and are collocated on chromosomes 1 and 8, detected from the same donor Akundi using two different F_2:3_ mapping populations derived from BR28/Akundi and BR49/Akundi (Table 3; Figure 6). Three genomic loci (*qSES1*, *qSL1*, and *qK1*) are located at the 151.8- 156.0 cM position on the bottom of chromosome 1. The other stable QTLs such as *qRL1* and *qSUR1* were found at 42.0–50.0 cM position of chromosome 1. However, chromosome 8 also harbored two collocated QTL regions (*qSL8* and *qK8*) that were positioned at 18.8–22.0 cM for governing shoot length and potassium absorption traits, respectively.

## 3. Discussion

The development and cultivation of salt-resilient rice cultivars are crucial strategies for enhancing rice yield and production and ensuring worldwide food security.

Salinity stress adversely effects rice plants by impeding seed germination, early seedling stage growth and development, and panicle initiation to flowering stages (reproductive phase) [15,21,22]. Moreover, the early seedling stage is important, and salinity tolerance at this stage plays a pivotal role for crop establishment in the coastal zones because soil salinity is generally high at the onset of the monsoon season [11]. The high sodium concentrations in saline soil limits water uptake and the absorption of nutrients in the plant [23]. Rice plants have to maintain the Na^+^/K^+^ homeostasis under salt stress by maintaining a high K^+^/Na^+^ ratio because excessive sodium (Na^+^) often leads to K^+^ deficiency [22].

Previous reports to dissect molecular mechanisms for salinity tolerance at the early seedling stage were regularly used for different salt-tolerant donors such as Nona Bokra, Pokkali, Hasawi, and Capsule. Limited attempts were made to find novel salt-tolerant germplasm, new QTL, and analyze digenic interaction for salinity tolerance. We used a Bangladeshi landrace Akundi as a donor in the present investigation. This landrace shows high resilience to salt stress, and it was collected from the coastal hotspot saline areas of Bangladesh [15]. BRRI dhan28, a highly popular *boro* mega rice cultivar (this single popular rice variety is cultivated by farmers that cover at least one million hectares of land or more areas) in Bangladesh, was also used as a parent that is sensitive to salt stress. The current study reports 12 QTLs, which were identified on seven different chromosomes controlling various component traits pertaining to salt tolerance. Multiple QTLs were collocated and found on chromosome 1 (*qSES1, qSL1, qSDW1, qRL1, qK1*), on chromosome 10 (*qSDW10, qNa10*), and the other QTL loci are identified on chromosome 2 (*qNa2*), on chromosome 7 (*qSPAD7*), on chromosome 8 (*qSL8*), on chromosome 11 (*qSUR11*), and on chromosome 12 (*qNaKR12*) through the QGene and ICIM-ADD Version 4.2 software [24,25]. Here, 12 putative candidate genes were identified for these QTLs, where their position, the total number of loci, and putative function are also mentioned. This study also identified 28 significant epistasis interactions by using ICIM-EPI Version 4.2 software. Seven QTLs such as *qSES1, qSL1, qRL1*, *qSUR1, qSL8, qK8,* and *qK1* were identified on chromosomes 1 and 8, detected using same donor Akundi but two different F_2:3_ mapping populations derived from BR28/Akundi and BR49/Akundi. These genetic loci are reliable, stable, and robust despite the genetic backgrounds being different (Table 3).

### 3.1. Assessing the Salt Stress Responses in the Selected F_2:3_ Progenies with Their Parents and Determining Cause and Effect Relationships via Path Analysis

The leading traits SES, SPAD value, Na^+^ and K^+^ concentration, Na^+^/K^+^ ratio, % survival, shoot length and root length, and shoot biomass have a vital role in salt resilience. In this study, SES scores of the tolerant parent (Akundi) ranged from 3 to 4 with an average of 3.5, but in sensitive parent SES, the score was 6 to 8 with an average of 7. The Na^+^ concentration in the sensitive parent was almost more than 0.31 mmolg^−1^dwt, whereas the tolerant parent had 0.18 mmolg^−1^dwt and Na^+^/K^+^ ratio was also higher in salt-sensitive individuals and lower Na^+^/K^+^ ratio was observed in salt-tolerant progenies. If the amount of Na^+^ in the plant is high, water absorption is lower and plant growth is slowed down. This is because the plant cell’s enzymatic connection is disrupted, which causes the plant to die early and reduces grain yield [7,11,26,27]. The tolerant plants had a high percentage of survival (100%), but a lower percentage of survival (45%) was found in the sensitive set. In this study, the tolerant set had a high shoot length of 57.2 cm, which ranged from 55.8 cm to 58.6 cm, while the sensitive set had a shorter shoot length of 28.0 cm. The tolerant set also had more shoot biomass (0.21 g), while the sensitive set had less (0.08 g), which strongly supports that higher biomass and shoot length accelerate the growth at the seedling stage, which lowers the Na^+^ concentration in the plant tissue [7,11,27].

The negative correlation was found among the SES score with K^+^ concentration, root length, and SPAD value; shoot and root length with Na^+^ concentration; and Na^+^/K^+^ ratio with K^+^ concentration, survival, shoot and root length. In salt stress conditions, a large amount of Na^+^ accumulated in the plant tissue causes severe cell damage; consequently, early plant death was observed and grain yield reduction also occurred [7,27]. The correlation between the different variables (traits) helps us in the selection of progenies in the plant breeding program [28]. The positive and significant association between SES score and Na^+^ concentration impacts the overall performance (phenotypic responses), as reported before [7].

On the basis of path analysis, different independent salt tolerance traits such as Na^+^ concentration, K^+^ concentration and Na^+^/K^+^ ratio, % survival, shoot length, shoot dry weight, root length, and SPAD value contributed to the overall salt tolerance. Therefore, the total salt tolerance is the result of these eight traits and has some residual effect (R), which agreed with the previous reports [7,27]. In this study, path coefficient analysis revealed that % survival, Na^+^ concentration, K^+^ concentration, and Na^+^/K^+^ ratio are the major traits that contribute to salt tolerance because these variables jointly explain a major phenotypic variation. Many researchers also made similar observations [7,11,29]. This study is strongly supported by those researchers. Therefore, for the breeding program for developing a salt-tolerant cultivar, we need to select the promising advanced lines based on these four characters, and path coefficient can play a great role in stress breeding, including breeding for salinity tolerance.

### 3.2. Marker Segregation and Important Salt-Tolerant QTL Regions

A total of 12 QTLs were identified in the current study, which are tolerant to salt conditions associated with agronomic and physiological traits. Five QTLs (*qSES1, qSL1, qK1* on chromosome 1 and *qSDW10* and *qNa10* on chromosome 10) described over 65.6% and 32.6% of the combined phenotypic variations, respectively. Current focus should be placed on these QTLs for extensive analysis to find better insights regarding their vital roles in regulating salt tolerance via the physiological process [7]. In the recent study, seven common QTLs (*qSES1, qSL1, qRL1, qSUR1, qSL8, qK8* and *qK1*) were reported both in QGene and ICIM-ADD Version 4.2 software. Therefore, these QTLs may be considered as main-effect QTLs in our study.

Multi-locus analysis shows the pooled effect of major QTLs *qSES1, qSL1, qK1* on chromosome 1 (pooled/combined PVE 65.6% of three QTL with R^2^ 15.6%, 30.7%, and 19.3%, respectively) and similarly *qSDW10* and *qNa10* on chromosome 10 (combined PVE 32.6% with R^2^ 17.1% and 15.5%, respectively; when the main QTL of these characters were considered simultaneously, their summation of single/individual effect is greater than their pooled effect. This is as a consequence of: (i) QTL co-localization or pleiotropy; (ii) additive epistatic interaction among QTLs is high; (iii) similar pathways/mechanisms are used for affecting this trait by some of the QTL.

### 3.3. Comparing the QTLs Revealed in the Present Study with Previously Reported QTLs

The QTLs identified in the present study were checked with the previously reported QTLs from various studies. In the early vegetative (seedling) stage, a main QTL (*Saltol*) for salt tolerance was detected, which is located on chromosome 1 and had a large phenotypic variation (R^2^ of 39.0–44.0%) from the traditional widely studied landrace Pokkali [30,31]. In addition, a main effect salt-tolerant gene (*SKC1*) responsible for maintaining K^+^/Na^+^ ionic balance (homeostasis) in response to salt stress is reported on chromosome 1 during the early vegetative stage from the Nona Bokra [32]. The QTLs/genes such as *Saltol* (10.5–11.5 Mb [30]), *OsSalT* [33], sal*T* [34], and *SKC1* (11.46 Mb [32]) were not found in Akundi. Three new QTLs (*qSES1*, *qSL1* and *qK1*) are co-located on chromosome 1, and two QTLs (*qSDW10* and *qNa10*) were identified on chromosome 10. and position was 68.5 cM. Therefore, these new QTLs are strongly responsible for tolerance to salt stress.

High-affinity potassium transporter (*HKT1*) has a role in excluding sodium and maintaining a high K^+^/Na^+^ ratio in leaves when imposed to salinity stress [35]. Previous investigations have shown that the maintenance of a low Na^+^ concentration in leaves is an important mechanism for rice plants to improve their salt stress tolerance [11,13,26,36,37]. ZmHKT1 causes leaf Na^+^ exclusion promotion and is identified as a major salt tolerance quantitative trait locus (QTL) [38]. The present study identifies two QTLs (*qNa2* and *qNa10*) on chromosomes 2 and 10, respectively, as exhibiting salinity tolerance through Na^+^ exclusion ability. The salt stress response is regulated by the circadian clock in plants. Several proteins that maintain the circadian clock play key roles in regulating salt stress tolerance [39,40]. Recently, studies have demonstrated a new molecular link between clock components and salt stress tolerance in rice. *Oryza sativa* pseudo-response regulator (*OsPRR73*) is induced by salt and specifically confers salt tolerance by recruiting HDAC10 to transcriptionally repress *OsHKT2;1*; therefore, it regulates rice salt tolerance [41]. Membrane compartment-localized aquaporins might also participate in ion homeostasis regulation through controlling root water uptake, leaf water transpiration, stomatal closure, and small molecule transport in response to salt stress. For instance, the overexpression of the wheat aquaporin TdPIP2;1 improves salt stress tolerance through retaining a low Na^+^/K^+^ ratio under high salt stress conditions [42]. However, one QTL (*qNaKR12*) was identified on chromosome 12 that maintains a low Na^+^/K^+^ ratio under salinity conditions in the current investigation. The position of the QTL is 39.4 cM, and the additive effect is −0.04, which explained the 12.1% phenotypic variation.

We summarized that for the salinity-tolerant QTLs from the previous studies in comparison to the present investigation, the QTL position for this study is totally different from the other studies [7,27,43,44] (Appendix A). Therefore, from this comparison, the present study suggests that all of these QTLs did not locate in the same position of previously reported well-known QTLs/gene such as *Saltol* [30], *SKC1* [32], *OsSalT* [33] (a gene controlling salt stress found on *Saltol* QTL region of rice), and sal*T* (a cDNA clone) was reported as salt stress-induced protein changes in the roots of the salt-sensitive *Oryza sativa*, var. *Indica,* cv Taichung native 1 [34]. Therefore, the QTLs identified in this study are novel, which could be used in rice breeding for enhancing salt tolerance.

Several QTLs had large effects, which are located on the chromosomes 1, 8, and 10. There was one QTL (*qSES1*) for SES score where LOD is 3.4 and R^2^ (PVE) of 15.6%; other QTLs are *qSL1* (LOD =7.3, R^2^ = 30.7%), *qK1* (LOD = 4.3, R^2^ = 19.3%), *qSDW10* (LOD = 3.6, R^2^ = 17.1%), and *qNa10* (LOD = 3.3, R^2^ = 15.5%). These two sets (*qSES1*, *qSL1*, *qK1* co-located on chromosome1 and *qSDW10*, *qNa10* shared common location on chromosome 10) of QTLs are co-located and have a functional kinship. Thus, these main effect QTLs may have pleiotropic effects (number of traits impacted by same QTL) on other traits. The QTLs sharing a common region located on chromosome 1 and chromosome 10 for component salt tolerance traits also supported that there is strong correlation among the traits.

From the above summary, the seven stable major QTLs were found from this study, which might be potential targets for QTL stacking and marker-enabled selection. Here, 12 QTLs were reported as novel QTLs that might be used for further breeding programs in rice for enhancing salt resilience.

## 4. Materials and Methods

### 4.1. Parent Selection

Akundi and BRRI dhan28 were chosen as parents to produce the bi-parental mapping population. Akundi is a Bangladesh-origin indica germplasm that shows salt tolerance at the early vegetative phase and was collected from southern coastal areas of Bangladesh. This region is rich in diverse salinity-tolerant germplasms. The agronomic features of Akundi include: (i) plant height of ~155 cml (ii) it has moderate sensitivity to photoperiod; (iii) produces robust seedlings with wide, long, and droopy leaves; (iv) moderate tillering ability, and panicles are approximately 20–25 cm long. Usually, the grains are bold and awnless, and their appearance is reddish-colored; (v) it has a low yield potential (1.9–2.5 tha^−1^) and matures within 120–125 days of growth duration [15].

BRRI dhan28 is a common indica irrigated (*boro*) rice type with 90–95 cm in height, typically awnless, and grains are medium slender; it matures in 140–145 days and gives yield 6.0–6.5 tha^−1^. BRRI dhan28 is a salt-stress-sensitive variety at the vegetative stage, but is photoperiod insensitive [45].

One F_2:3_ family derived from the cross of BRRI dhan28/Akundi was used in the present study for genetic dissection of traits linked with salinity tolerance.

The present investigation also compared genetic loci detected from two mapping populations derived from the cross of landrace Akundi with BRRI dhan49 [7] and BRRI dhan28/Akundi.

### 4.2. Growing Conditions

The experiments were performed at the Bangladesh Rice Research Institute (BRRI; http://brri.gov.bd/; accessed on 1 May 2023), Gazipur 1701, Bangladesh, during the irrigated/dry (*boro*) season. Two parents (BRRI dhan28 and Akundi) were grown in three different sets. Seeds of the first set were sown on 15 November 2019, the second set was seeded with a 7-day interval from the first set, and the third set was sown 7-days later to match flowering times to make a successful cross through hybridization. The thirty-day-old single seedling was transplanted in a 5.0 m × 8 rows plot with a spacing of 20 × 20 cm. Fertilizer doses were 80, 60, 40, and 20 kg NPKS/ha, with nitrogen applied in three stages (40 + 20 + 20). The total amount of P, K, and S was utilized during the final plot/land preparation. Additional measures were taken as and when necessary. Eventually, the crossing was done in the net house of the Plant Breeding Division at the end of the season, and F_1_ seeds were harvested ~25 days after pollination. The F_1_ plants were confirmed as a true hybrid using the 10-SNP panel. Next, F_2_ seeds were harvested and collected with proper labeling, and then preserved from selected crosses. In the following rainfed lowland rice (RLR/wet) season, on 7 July 2020, F_2_ seeds were sown in the seedbed. A twenty-day-old single seedling of F_2_ individuals was transplanted in the field. After seedling establishment in the field, the leaf samples were collected for genotyping at 25 days after sowing. In 2021, the F_2:3_ seedlings were screened and phenotyped in hydroponics in a phytotron that was set at 30 °C/22 °C day/night temperature, respectively, and 65–70% relative humidity was maintained. To initiate germination, seeds were oven dried for 3 days at 50 °C, then surface sterilized using a fungicide (Vitavax-200, Syngenta) and washed with distilled water. Seeds were then placed in petri dishes lined with moistened filter paper and incubated at 30 °C for 48 h. Two pre-germinated seeds were sown per hole in 10 L plastic trays using Styrofoam seedling floats floating in distilled water for three days, followed by a Yoshida culture solution [46]. Salt stress was imposed 14 days after sowing by adding salt (NaCl) to the culture solution until an electrical conductivity of 12 dSm^−1^ was achieved. Sodium metasilicate 9 hydrate (4.5 mg L^−1^) was applied as a source of silicon to avoid the lodging of the plants. The culture solution was made acidic every day to prevent iron (Fe) deficiency by maintaining a pH of 5.0 and replaced the culture solution at every seven days interval.

### 4.3. Characterizing Agronomic Traits

For describing specific morphological and physiological features, 92 F_2:3_ plants were characterized that developed from a cross between BRRI dhan28 and Akundi. The seedling salt stress injury symptoms were phenotypically assessed using SES scores [47], with a magnitude of one indicating high tolerance to salt stress and a score of 9 suggesting highly sensitive genotypes. Sampling was done from individual plants 21 days after salinization, dried for 3 days at 70 °C, and then the dried samples were weighed. Sodium—potassium concentrations were then measured [12,15,48,49]. Survivors were recorded at three weeks after applying salt stress, and the % survival was calculated using the number of seedlings during the initiation of the salt treatment. The shoot length was estimated by measuring the distance between the stem base and the tip of the longest leaf. The individual plant was harvested, including all plant components and the root. All samples were oven dried for three days at 70 °C before being weighed. The root length was determined from the base of the stem to the longest root tip.

### 4.4. Determining Physiological Traits Response

The individual plant was taken from different F_2:3_ families and washed three times using deionized water prior to drying. Each plant was sun-dried for three days before drying further at 50 °C in an oven for three days. After drying, the plant material was ground and weighed and about 0.50 g powdered sample was kept in a test tube where 25 mL 1N HCl was poured before. The digested samples were filtered after 24 h of digestion in 1N HCl. The one ml extract was then diluted by adding 39 mL 1N HCL. Then, a reference solution was prepared, and a flame photometer (Model410) was used to analyze sodium and potassium concentrations [46].

### 4.5. SPAD Reading

The chlorophyll content was estimated from five entirely expanded third leaves selected from each replicate using a SPAD meter (Minolta 502, Osaka, Japan). The leaves’ chlorophyll content was determined using a non-destructive method before harvesting.

### 4.6. Analyzing Trait Associations (Correlation)

A total of 92 F_2:3_ individuals were carefully chosen for this investigation from the cross of BRRI dhan28/Akundi. The strength of the relationship measures through correlation coefficients comprising different characters was assessed using the RStudio 4.1.1 software [50].

### 4.7. Path Coefficient Analysis

Path coefficient analysis [51,52] was used to calculate the direct and indirect contributions/effects of various traits (independent variables) to SES (dependent variable). The effect of the remainder variables that are not under-studied is estimated as the residual effect.

### 4.8. SNP Genotyping and Genetic Linkage Map Construction

Leaf samples of three-week-old plants were collected for DNA extraction from 92 F_2_ individuals derived from the cross combination of BRRI dhan28/Akundi. Then, leaf samples were preserved at an extremely low temperature of −80 °C for 1k-RiCA genotyping. The 1k-RiCA (1K Rice Custom Amplicon) assay was constructed on Illumina’s TruSeq Custom Amplicon (TSCA) 384 Index Kit technology (https://www.illumina.com; accessed on 21 October 2022) using Illumina’s registered workflow [53]. During sample preparation, each leaf sample was perforated into a small piece and placed into a specific well of a 96-well plate based on the exact sequence of a sample. Only one PCR well plate was needed for keeping 92 individual samples. Then, the sample plate was put into an oven for drying samples at 50 °C for 24 h and bundled in a zipper bag. Finally, the 92 oven-dried samples were sent for 1k-RiCA SNP assay for the whole genome genotyping using 945 SNP markers through genotyping by sequencing technique. The 1k-RiCA SNP assay was done at the Agriplex genomics, Cedar Avenue, Suite 250, Cleveland, 011,444,106, USA. The 945 markers’ molecular linkage map was constructed using the Nipponbare genome sequence (https://www.gramene.org/; accessed on 23 June 2022), and the SNP marker physical distances were calculated by multiplying the megabase pair (Mb) positions by 3.924 to get a corresponding estimate of centiMorgan (cM; 1 cM = ~ 244 kb [53]).

### 4.9. QTL Dissection

Molecular marker assay and genetic mapping were done on the 92 F_2_ individuals selected from the cross BRRI dhan28/Akundi. These plants were genotyped using 886 SNP markers, and 886 bin markers with genome-wide coverage of the BRRI dhan28/Akundi population were employed to build a genetic linkage map. The genetic map is spanned by 1.72 cM on the average distance between two SNPs. To estimate the association between specific SNP marker loci and phenotypic traits linked with salinity tolerance, QTL dissection was carried out using QGene 4.0 [54] and the ICIM-ADD method (QTL IciMapping Version 4.2 software [24,25,55]). Single marker regression (SMR) analysis, interval mapping (IM), and composite interval mapping (CIM) were applied to detect the position of the revealed QTL for salinity tolerance. The minimum LOD value is estimated to declare a QTL significant. QTLs were determined by the CIM method using permutation analysis at 1000 iterations [56]. The value of R^2^ (PVE: phenotypic variation explained by the QTL) was calculated as the portion of the total PVE by each QTL. The forward cofactor selection method was used for QTL detection in the CIM method using QGene. The additive effects and contributor of favorable allele/QTL for each trait of interest were calculated.

### 4.10. Epistatic QTLs Identification

Epistatic QTLs were identified using the epistasis mapping method ICIM-Epi [57] using the current population. Next, two-dimensional scanning (or interval mapping) was performed to detect significant epistatic QTLs using the multiple regression model. The LOD threshold for epistasis mapping was 5.0. Three possible digenic epistasis can be identified: (i) QTL vs. QTL (both QTLs may have additive effects; (ii) QTL vs. background loci; (iii) between complementary loci (background loci vs. background loci).

## 5. Conclusions

Salinity tolerance is complex abiotic stress that is controlled by multigenes. Here, the contribution of traits for salt tolerance was assessed through correlation and path analysis. The results suggested that if Na^+^ concentration and Na^+^/K^+^ ratio increased in the plant tissue, then the early death of rice occurs. Therefore, % survival, Na^+^ concentration, K^+^ concentration, and Na^+^/K^+^ ratio are the key mechanisms in the plant tissue at the seedling stage. The present investigation reported two biparental mapping populations developed from the cross of Bangladeshi landrace Akundi with two sensitive parents BRRI dhan28 and BRRI dhan49. The study detected seven large-effect QTLs for salt tolerance across the populations. Here, 12 QTLs were identified by single marker regression, interval mapping, and composite interval mapping method using QGene and ICIM-ADD Version 4.2 software. The QTLs identified on chromosome 1 (*qSES1, qSL1, qSDW1, qRL1, qK1*), 2(*qNa2*), 7(*qSPAD7*), 8(*qSL8*), 10 (*qSDW10*, *qNa10*), 11 (*qSUR11*), 12 (*qNaKR12*) are novel because these QTLs do not match with other genetic loci that have been previously identified. In this study, five major QTLs (*qSES1*, *qSL1*, *qK1*, *qSDW10* and *qNa10*) were identified and they have a major effect on salt tolerance. Seven salinity-resilient QTLs (SRQ: *qSES1*, *qSL1*, *qRL1*, *qSUR1*, *qSL8*, *qK8*, *qK1*) were found to be stable in the different genetic backgrounds (genetic background independent) that co-exist on chromosomes 1 and 8, detected from the same donor Akundi using two different F_2:3_ mapping populations derived from BR28/Akundi and BR49/Akundi. These stable QTLs may be important for pyramiding and marker-enabled selection in future breeding programs, and will validate the introgression of these QTL regions into advanced breeding lines developed from both studies. These QTLs will help to develop rice cultivars that can withstand salt stress. These will be good targets for extensive study through QTL fine mapping and QTL deployment for salt tolerance breeding.

## Figures and Tables

**Figure 1 ijms-24-11141-f001:**
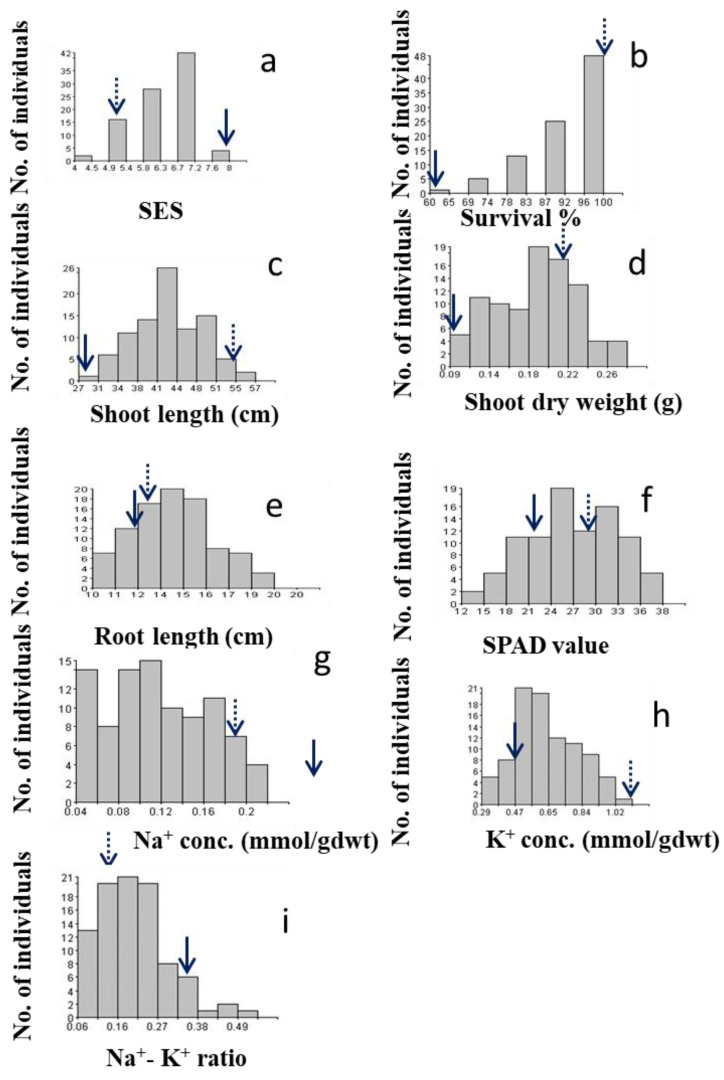
Histogram shows frequency distribution (no. of individuals) of the F_2:3_ populations for different traits related to salinity tolerance during the early seedling stage; (**a**) Salt injury score (SES score), (**b**) % Survival, (**c**) Shoot length (cm), (**d**) Shoot dry weight (g), (**e**) Root length (cm), (**f**) SPAD value, (**g**) Sodium concentration (Na-Conc.), (**h**) Potassium concentration (K-Conc.), (**i**) Na^+^/K^+^ ratio. Dotted and solid arrows on the rectangle show the trait value of the salt-tolerant and sensitive parent Akundi and BR28, respectively.

**Figure 2 ijms-24-11141-f002:**
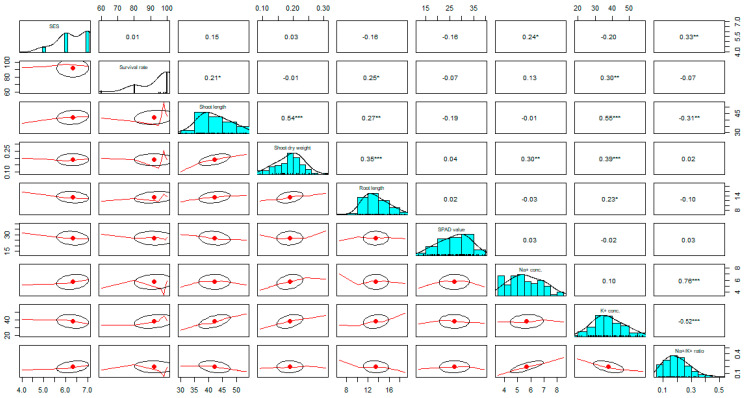
Strength of the relationship among different component traits associated with salinity tolerance was measured using correlation analysis in an F_2:3_ population developed from a cross BR28 (sensitive to salt stress)/Akundi (salt tolerant) at the seedling stage. The higher value of correlation (r) denotes strong association between two variables and the lower value of correlation indicates weak correlation. Conc.: Concentration. *, **, *** significant at *p* < 0.05, *p* < 0.01 and *p* < 0.001 respectively. Tabulated t-value at *p* < 0.05 = 0.204, *p* < 0.01 = 0.267 *p* < 0.001= 0.337.

**Figure 3 ijms-24-11141-f003:**
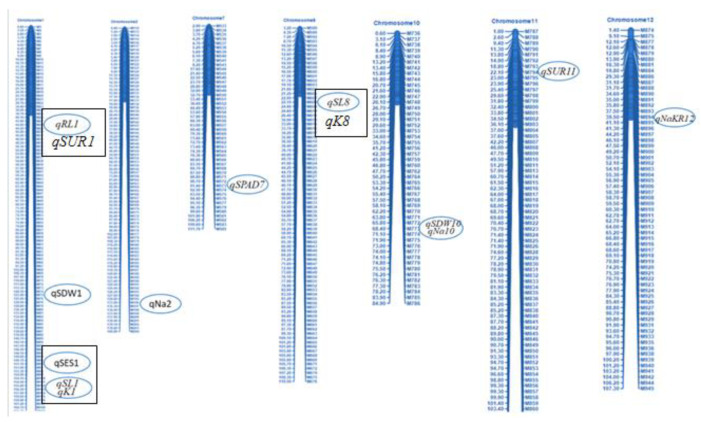
A molecular marker-based linkage map of *Oryza sativa* L. rice F_2:3_ population generated from a cross between BR28 and Akundi with the help of QTL IciMapping Version 4.2 software. A total of 886 SNP markers were assigned to detect 12 QTLs on seven linkage groups under salinity stress of EC 12 dSm^−1^ at the seedling stage of rice. The designation of the SNP IDs is illustrated on the right, and the elliptical boxes adjacent to SNP markers denote the approximate positions of the 12 detected QTLs for salt tolerance in BR28/Akundi population. QTLs which are shown in rectangular boxes indicating stable and common QTLs (*qSES1*, *qSL1*, *qRL1*, *qK1*, and *qSL8*; see Table 3) observed in both BR28/Akundi and BR49/Akundi populations. However, *qK8* (from BR49/Akundi population) share similar positions with *qSL8* (from BR28/Akundi population) and *qSUR1* (BR49/Akundi population) located close to *qRL1* (BR28/Akundi population).

**Figure 4 ijms-24-11141-f004:**
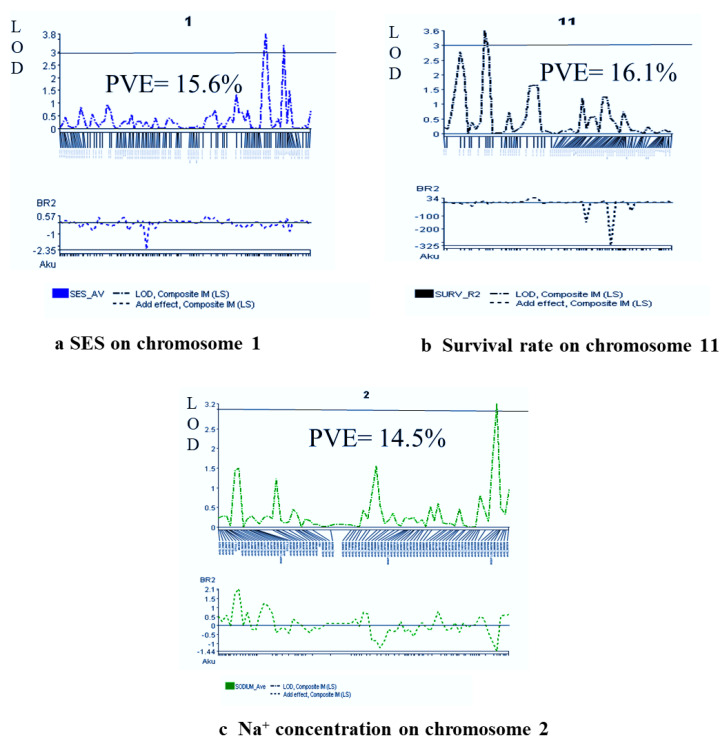
QTL probability graphs of the LOD score (significance threshold of LOD = 3.0) of different traits ((**a**) SES Score, (**b**) % Survival, and (**c**) Na^+^ concentration) indicate the strength/power of evidence for the occurrence/presence of a QTL on a particular chromosomal location.

**Figure 5 ijms-24-11141-f005:**
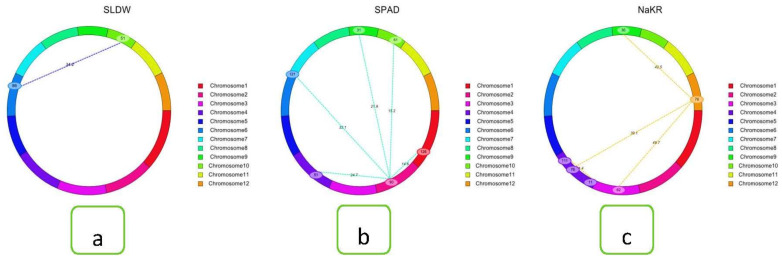
Cyclic diagrams of epistatic QTLs for various salt tolerance traits: (**a**) shoot dry weight, (**b**) SPAD value, and (**c**) Na^+^/K^+^ ratio. The spotted–marked lines designate the interacting marker pairs found on the same or different chromosomes with respective LOD values due to epistatic effect.

**Figure 6 ijms-24-11141-f006:**
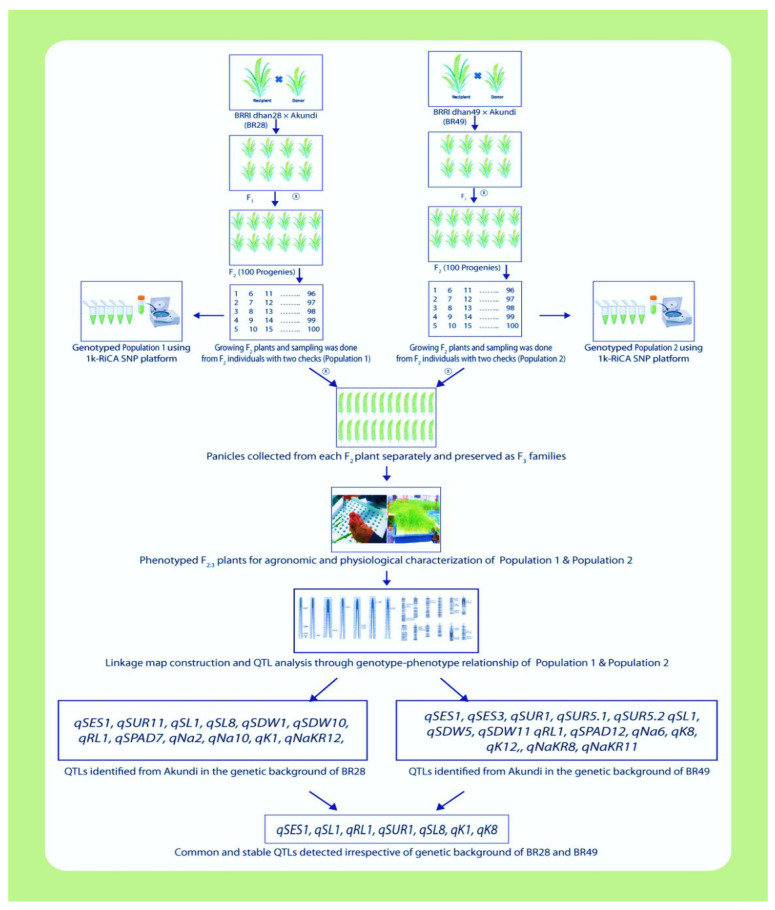
Schematic illustration for molecular mapping by 1k-RiCA SNP genotyping assay using two different bi-parental mapping populations to discover reliable and stable QTLs for salt resilience under salt stress in rice (partly designed with BioRender.com, accessed on 28 August 2022).

**Table 1 ijms-24-11141-t001:** Trait association (correlation) and path coefficients for estimating the contribution of studied traits through analyzing the possible causal relationship/linkage between independent (causal) variables (different traits such as % survival, shoot length, etc.) on dependent variables (effect) such as overall phenotypic performance (SES scores) under salt stress.

Variable Name	Correlation	% Survival	Shoot		Root Length	SPAD Value	Na^+^ Concentration	K^+^ Concentration	Na^+^/K^+^ Ratio	Total Effect
			Length	Dry Weight						
% Survival	0.008	0.035	0.093	0.001	−0.042	0.006	0.021	−0.096	−0.011	0.008
Shoot length	0.147	0.007	**0.440**	−0.046	−0.044	0.016	−0.001	−0.173	−0.052	0.147
Shoot dry weight	0.018	0.001	0.235	**−0.086**	−0.058	−0.004	0.049	−0.121	0.003	0.018
Root length	−0.161	0.009	0.118	−0.030	**−0.164**	−0.001	−0.004	−0.072	−0.017	−0.161
SPAD value	−0.162	−0.003	−0.083	−0.004	−0.003	**−0.086**	0.005	0.005	0.006	−0.162
Na^+^ conc.	0.239	0.004	−0.002	−0.025	0.004	−0.003	**0.164**	−0.031	0.127	0.239
K^+^ conc.	−0.201	0.011	0.243	−0.033	−0.038	0.001	0.016	**−0.314**	−0.087	−0.201
Na^+^/K^+^ ratio	0.328	−0.002	−0.137	−0.001	0.016	−0.003	0.125	0.163	**0.168**	0.328

Residual effect: 0.27. Direct effects (diagonal values shown in bold) of different studied traits on overall phenotypic performance (SES scores) under salt stress.

**Table 2 ijms-24-11141-t002:** QTLs identified for quantitative characters associated with salt tolerance during the seedling stage in BR28/Akundi F_2:3_ population.

Characters	QTL Identified	Chr.	QTL Peak Marker	QTL Position (cM)		Additive Effect	LOD		PVE (%)		QTL Detection Method	Favorable Allele Contributing Parent
				QGene	ICIM		QGene	ICIM	QGene	ICIM		
SES	*qSES1*	1	chr01_38632196	151.8	150.8	−0.61	3.4	3.0	15.6	2.59	IM, CIM	Akundi
% Survival	*qSUR11*	11	chr11_5615885	21	21	13.73	3.5	4.0	16.1	0.85	IM, CIM	BR28
Shoot length	*qSL1*	1	QSES1-2_2	156	-	4.86	7.3	-	30.7	-	IM, CIM	BR28
	*qSL8*	8	GM4_4	22	-	4.29	3.2	-	16.3	-	IM, CIM	BR28
Shoot dry weight	*qSDW1*	1	chr01_231396842	123.4	137.8	−0.25	3.6	6.0	16.4	13.44	IM, CIM	Akundi
	*qSDW10*	10	chr10_17397576	68.6	64.6	−0.2	3.6	5.0	17.1	17.31	IM, CIM	Akundi
Root length	*qRL1*	1	chr01_1045259	42	-	1.93	3.5	-	26	-	SMA	BR28
SPAD value	*qSPAD7*	7	AG3_1	91.4	-	−2.32	3.0	-	12.1	-	SMR	Akundi
Na^+^ conc.	*qNa2*	2	chr02_34072964	134.4	-	−1.43	3.1	-	14.5	-	IM, CIM	Akundi
	*qNa10*	10	chr10_17397576	68.8	68.6	−0.74	3.3	5.0	15.5	21.20	IM, CIM	Akundi
K^+^ conc.	*qK1*	1	QSES1-2_2	156.5	156.8	0.102	4.28	4.0	19.3	18.75	SMR	BR28
Na^+^/K^+^ ratio	*qNaKR12*	12	chr12_10051752	39.4	-	−0.04	3.0	-	12.1	-	CIM	Akundi

SMA = Single marker association, IM = Interval mapping, CIM = Composite interval mapping, SMR = Single marker regression.

**Table 3 ijms-24-11141-t003:** Common, reliable, and stable salinity-resilient QTLs (SRQ) sharing common chromosomal regions on two different chromosomes detected from the same donor Akundi using two different F_2:3_ mapping populations derived from BR28/Akundi and BR49/Akundi.

Trait	Population-1			Favorable Allele Contributing Parent	Trait	Population-2			Favorable Allele Contributing Parent	Common and Stable QTLs in Different Genetic Background
	QTL Detected in Pop. 1	Chr. No.	QTL Position (cM)			QTL Detected in Pop. 2	Chr. No.	QTL Position (cM)		
SES	*qSES1*	1	151.8	Akundi	SES	*qSES1*	1	151.8	Akundi	*qSES1*
Shoot length	*qSL1*	1	156	BR28	Shoot length	*qSL1*	1	156	BR49	*qSL1*
Root length	*qRL1*	1	42	BR28	Root length	*qRL1*	1	48.8	Akundi	*qRL1*
Root length	*qRL1*	1	42	BR28	Survival (%)	*qSUR1*	1	50	BR49	*qSUR1*
Shoot length	*qSL8*	8	22	BR28	K^+^ Conc.	*qK8*	8	18.8	Akundi	*qSL8*
Shoot length	*qSL8*	8	22	BR28	K^+^ Conc.	*qK8*	8	18.8	Akundi	*qK8*
K^+^ conc.	*qK1*	1	156.5	BR28	Shoot length	*qSL1*	1	156	BR49	*qK1*

## Data Availability

Not applicable.

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
