# Peer review of "Molecular Mapping to Discover Reliable Salinity-Resilient QTLs from the Novel Landrace Akundi in Two Bi-Parental Populations Using SNP-Based Genome-Wide Analysis in Rice"

_ijms, 2023, doi:10.3390/ijms241311141_

Round 1

Reviewer 1 Report

The paper of Maniruzzaman et al. conducted a genetic analysis of salinity tolerance using an Indica x Japonica populations. Overall, the paper presented only a preliminary result and didn't identify the causal gene/s responsible for the trait of interest. However, this could still serve as a reference for researchers who wanted to finemap the identified QTL in this study. As of the moment, I do not recommend publication of this manuscript until the authors address the following comments. 

1. In the introduction, updated references must be added on the previously identified salt tolerance QTL. In addition, the authors must include another paragraph about next-generation sequencing and its role in identification of QTL. You may refer to the manuscript I attached here to see the full details of my recommendation. Please also revise the last paragraph, make sure that it is concise and straight forward as to what is the objective of your study.

2. Across the entire manuscript, the authors must correct nomenclature of important information such as QTL name (must be italicized), F2:3,etc.

3. In the result part, there are several information that can be transferred to discussion part. In addition, several subclauses can be merge together. 

4. Please generate new image for Figure 4 and Figure 6 as they are not eye-friendly (names and information are too small making them almost useless).

5. Tables, please make all the tables uniform (same font size, etc). 

Additional comments can be found in the attached file. 

Author Response

Reviewer 1:

The paper of Maniruzzaman et al. conducted a genetic analysis of salinity tolerance using an Indica x Japonica population. Overall, the paper presented only a preliminary result and didn't identify the causal gene/s responsible for the trait of interest. However, this could still serve as a reference for researchers who wanted to fine map the identified QTL in this study. As of the moment, I do not recommend publication of this manuscript until the authors address the following comments. 

  1. In the introduction, updated references must be added on the previously identified salt tolerance QTL. In addition, the authors must include another paragraph about next-generation sequencing and its role in identification of QTL. You may refer to the manuscript I attached here to see the full details of my recommendation. Please also revise the last paragraph, make sure that it is concise and straight forward as to what is the objective of your study.

Response: Thanks for your suggestion. Introduction has updated in the manuscript based on your suggestions.

  1. Across the entire manuscript, the authors must correct nomenclature of important information such as QTL name (must be italicized), F2:3, etc.

Response:  Corrected QTL name (italicized) and F2:3, throughout the manuscript.

  1. In the result part, there are several information that can be transferred to discussion part. In addition, several subclauses can be merge together.

Response:  We have kept and depicted the results in the result section only. 

  1. Please generate new image for Figure 4 and Figure 6 as they are not eye-friendly (names and information are too small making them almost useless).

Response:  Figure 4 is software generated and both the images were enlarged to make more visible.

  1. Tables, please make all the tables uniform (same font size, etc). 

Response:  All the tables were made similar font size.

Additional comments can be found in the attached file. 

Reviewer 2 Report

A wide range of abiotic stresses adversely impact rice production and yield. Hence, a number of breeding programs are aimed to improved crop performance under severe climatic conditions. Here, the authors have perused to discover novel genetic loci that contributes salt tolerance in rice seedlings.

The study provides key insights and novel QTLs for salt resistance at early stage. The manuscript could be considered for publication after careful revision.

Some comments:

1.       Although abstract provides information on number of QTLs, author should revise and provide statistical information such p-value and standard error/deviations.

2.       The first paragraph of introduction needs to be elaborated. It is too brief and do not convey significant message.

3.       Line 70: delete suitable, desirable is enough to send clear message.

4.       Line 73: Change to ‘Na+/K+ ratio’; change it everywhere.

5.       Hierarchical numbering in results section could be avoided. Moreover, for results subheadings provide clear and meaningful ‘complete sentence’ titles

6.       Figure 2. the faint colors are invisible in white background. Please modify the figure to make it clear and visible.

7.       Table 3. Organize table to a single page.

8.       Please expand introduction and discussion on DNA markers for molecular breeding in other crops. For instance, see few examples from literature on molecular marker-based analysis (using RAPD, RFLP, AFLP, SSR, ISSR, ITS, etc): Multiplex molecular marker-assisted analysis of significant pathogens of cotton (Gossypium sp.), 2022; Biocatalysis and Agricultural Biotechnology https://doi.org/10.1016/j.bcab.2022.102557 (Cotton);  Assessment of genetic diversity and volatile content of commercially grown banana (Musa spp.) cultivars, Hinge et al., Scientific Reports, 2022; https://doi.org/10.1038/s41598-022-11992-1 (Banana); Microsatellite and RAPD analysis of grape (Vitis spp.) accessions and identification of duplicates/misnomers in germplasm collection, Upadhyay et al., 2010 Indian J Hortic Volume 67 Pages 8-15; Microsatellite analysis to differentiate clones of Thompson seedless grapevine, Upadhyay et al., 2010, Ind Journal of Horticulture, Volume 67 Issue 2 Pages 260-263

Language editing would be helpful.

Author Response

Reviewer-2

A wide range of abiotic stresses adversely impact rice production and yield. Hence, a number of breeding programs are aimed to improved crop performance under severe climatic conditions. Here, the authors have perused to discover novel genetic loci that contributes salt tolerance in rice seedlings.

The study provides key insights and novel QTLs for salt resistance at early stage. The manuscript could be considered for publication after careful revision.

Some comments:

  1. Although abstract provides information on number of QTLs, author should revise and provide statistical information such p-value and standard error/deviations.

Response: Thanks for your suggestion. For declaring significant QTL, LOD threshold 3 was used as significance threshold.

  1. The first paragraph of introduction needs to be elaborated. It is too brief and do not convey significant message.

  Response: The first paragraph of introduction elaborated accordingly.

  1. Line 70: delete suitable, desirable is enough to send clear message.

  Response: Deleted.

  1. Line 73: Change to ‘Na+/K+ratio’; change it everywhere.

Response: Thank you. We have changed throughout the manuscript.

  1. Hierarchical numbering in results section could be avoided. Moreover, for results subheadings provide clear and meaningful ‘complete sentence’ titles

Response: Hierarchical numbering in results section was done based on journal format.

  1. Figure 2. the faint colors are invisible in white background. Please modify the figure to make it clear and visible.

Response:  Colors of Figure 2 have improved and it was generated through R-program and the color intensity indicates the strength of correlation.

  1. Table 3. Organize table to a single page.

Response: Table 3 arranged and accommodated in one page.

Please expand introduction and discussion on DNA markers for molecular breeding in other crops. For instance, see few examples from literature on molecular marker-based analysis (using RAPD, RFLP, AFLP, SSR, ISSR, ITS, etc): Multiplex molecular marker-assisted analysis of significant pathogens of cotton (Gossypium sp.), 2022; Biocatalysis and Agricultural Biotechnology https://doi.org/10.1016/j.bcab.2022.102557 (Cotton);  Assessment of genetic diversity and volatile content of commercially grown banana (Musa spp.) cultivars, Hinge et al., Scientific Reports, 2022; https://doi.org/10.1038/s41598-022-11992-1 (Banana); Microsatellite and RAPD analysis of grape (Vitis spp.) accessions and identification of duplicates/misnomers in germplasm collection, Upadhyay et al., 2010 Indian J Hortic Volume 67 Pages 8-15; Microsatellite analysis to differentiate clones of Thompson seedless grapevine, Upadhyay et al., 2010, Ind Journal of Horticulture, Volume 67 Issue 2 Pages 260-263

   Response: Thanks for the suggestion. We have expanded the introduction and discussion using the information and references relevant to the content of the manuscript.

Round 2

Reviewer 1 Report

First sentence in the introduction. Please include scientific references that rice is a glycophyte. 

Other parts are ok.

Needs grammatical review.